# Not All Late Filers Are the Same: Distinguishing between Differences in Filing Behaviour

**Thomas Selleslagh** [1,*], **Stefanie Ceustermans** [1] **and Lara Stas** [2,3]

1 Department of Business, Vrije Universiteit Brussel, 1050 Ixelles, Belgium; stefanie.ceustermans@vub.be
2 Interfaculty Center for Data Processing and Statistics, Core Facility, Vrije Universiteit Brussel, 1050 Brussel, Belgium; lara.stas@vub.be
3 Biostatistics and Medical Informatics Research Group, Faculty of Medicine and Pharmacy, Department of Public Health, Vrije Universiteit Brussel, 1050 Brussel, Belgium
\* Correspondence: thomas.selleslagh@vub.be

**Abstract:** This paper investigates the association between private firms' timeliness of financial reporting and financial health by exploring firms' reporting behaviour over a longer period of time (9 years). We show that 9% of all firms are consistently late every year and find significant differences in the association between firms' financial health and late filing depending on their past filing behaviour. We find a negative association between firms' financial health and late filing. However, our research also shows the opposite (i.e., positive) association for firms who were late consistently. Our results suggest that other motivations besides obfuscating bad performance might cause firms to delay the disclosure of their financial statements.

**Keywords:** financial reporting; timeliness; late filings; reporting delays; private firms; financial health





## 1. Introduction

Accounting and financial reporting are important concepts within the economy and society in general. The European Parliament acknowledges this importance and calls for a deeper understanding of the macroeconomic effects of accounting standards and the impact on long-term financial stability (European Parliament resolution (2018/C 086/03). Financial reporting is an important element to consider when aiming for sustainable development and assuring the health of the economy [1]. Within the domains of financial reporting and accounting, timeliness is an important issue. It refers to the amount of time it takes to disclose financial statements. The timing of disclosure is especially relevant to the stakeholders. Information must be made available to the users early enough to help them make decisions. The more time that elapses between the closing of a firms' accounting period and the disclosing of their financial statements, the less useful they are in the decision process of their stakeholders. In other words, the time it takes to disclose information is inversely related to the usefulness of that information.

The importance of timeliness is acknowledged by standard setters, policy makers, and legislators alike. Timeliness is one of the qualitative characteristics set out in the Conceptual Framework for Financial Reporting by The International Accounting Standards Board (IASB). According to those characteristics, information should be relevant, meaning that it should be capable of making a difference in making decisions. Under the new Accounting Directive (2013/34/EU), Europe also wants to stimulate timely financial reporting. Member States must ensure that private firms publish their financial statements "within a reasonable period of time, which shall not exceed 12 months after the balance sheet date" (DIRECTIVE 2013/34/EU). The idea behind compulsory and timely disclosure of financial statements, as described in the accounting directive set out by the European Commission, is to make "company information more easily and rapidly accessible by interested parties" (DIRECTIVE 2003/58/EC). In other words, the objective is to reduce

information asymmetries by improving the access but also the speed at which financial statements are disclosed. This is especially relevant in the context of private firms, which are typically more informationally opaque compared to larger listed firms (see e.g., Berger and Udell [2]).

Disclosure literature has shown that there are costs and benefits to disclosing information [3,4]. Firms will try to manage these costs and benefits to find an optimal disclosure strategy [4,5]. Previous research on disclosure timeliness of private firms has shown that late filings are no anomaly and are, in fact, very common [6–8]. There exist different explanations as to why firms would choose to delay the disclosure of their financial statements. Firms might be reluctant to share sensitive information due to competitive reasons [8,9]. Alternatively, following the agency theory, managers might also have incentives to obfuscate bad performance from investors. Despite these different reasons, most previous empirical research on the subject tends to focus primarily on the assumption that "bad news travels slow". Consequently, one of the main findings of previous studies is that late filings are associated with financial distress [8,10–12]. Furthermore, most studies on the timeliness of financial reporting have only been carried out on a relatively short time frame. Hence, these studies have not considered the impact of multiple late filings. Luypaert et al. [6] acknowledge the current lack of empirical research on this topic and call for further research on multiple late filings. Up to now, much less attention has been paid to the notion that late filing might as well be part of a firm's long term disclosure strategy, rather than an ad hoc decision to delay the disclosure of unfavorable information. By looking at a longer time frame, we are able to take into account a firm's past filing behaviour. Luypaert et al. [6] found that past filing behaviour is the most important predictor of a firm's current filing behaviour. However, the time periods covered in their study encompass only two consecutive years. Given this limitation, it is unclear whether the relationship between filing lags and financial health is the same for all firms.

The aim of this paper was to investigate whether consistent late filings in the previous years moderate the relationship between late filing in the subsequent year and a firms' financial health. We employ a large sample of 103,986 unique private firms covering a recent economic period (2010–2020).

This paper contributes to the literature in several ways. First, we add to the existing body of literature examining the importance of accounting within the private firm context. Despite their economic significance, still relatively little is known about their filing behaviour. Second, by employing a longer time frame of nine years, this is the first paper to report on multiple late filings. The findings reported shed new light on late filing and the association with firms' financial health. In reviewing the literature, no data was found on the association between the number of late filings and financial health. Our study is the first to show that the association between filing behaviour and firms' financial health is moderated by consistent late filings. Considering the efforts of the European Commission to improve the speed at which company information is made accessible and the European Parliament's call for a greater understanding of accounting standards, our findings will be of interest to policy makers and legislators as well. Considering the significance of financial reporting for the health of the economy and sustainable development [1], it also relates to the Europe 2020 strategy for smart, sustainable and inclusive growth; in fact, it relates even more so when taking into account that we focus on the relation between timeliness of financial reporting and firms' health, as firms' health is unmistakably a crucial aspect of long-term sustainable growth. That is to say, a firm's financial performance is regulated by the relationship with its key stakeholders and its reputation [13–17]. The transparency and reporting practices of a firm help in shaping this reputation [18].

The rest of this paper is structured as follows: in Section 2 we go over the relevant prior literature and formulate our hypotheses. Section 3 discusses our data and research design. Sections 4 and 5 present our results. Section 6 provides the results of our post hoc analyses. Section 7 provides a discussion. Section 8 concludes.

## 2. Literature Review and Hypothesis Development

From the perspective of the stakeholder of the firm, there are clear benefits from the (timely) disclosure of financial statements, including increased transparency and reduced information asymmetry. From the perspective of the disclosing company, however, the relation is not that straightforward, and there will exist different incentives to delay the disclosure of financial statements. These incentives can be roughly divided into two categories and are related to the financial disclosure theory of Darrough and Stoughton [3]. On the one hand, there is the obfuscation theory which, to put it simply, states that bad news travels slower than good news and can be linked to the agency effect and asymmetric information. This was established theoretically by Dye and Sridhar [19]. The management of firms with unfavorable results are more likely to delay the disclosure of their financial statements to obfuscate their bad performance and related agency problems. In previous literature, this has been empirically validated on numerous occasions (e.g., Begley and Fischer [20], Givoly and Palmon [21], Whittred [22], Haw et al. [23]).

On the other hand, there is the proprietary cost theory. Competitors might use the information contained in financial statements to their advantage. In other fields of literature (e.g., strategic management accounting), this is referred to as 'competitive accounting'. Previous literature not only indicates that financial statements are an important source of information for firms analyzing their rivals in the sector, but that employing this information contributes significantly to these firms' competitive advantage and profitability [24,25]. Hence, firms might be reluctant to disclose this information on time, as it can be a source of competitive advantage for their rivals. This has also been shown by Graham et al. [26], who concludes that fear of giving up proprietary information is one of the main barriers to voluntary disclosure. Graham et al. [26] further show that this fear is more pronounced for private firms. Accordingly, some firms might be willing to pay a premium (i.e., a penalty) to delay the publication of their valuable information. Furthermore, as pointed out in the dissertation of Wittmann [9], other stakeholders could exploit financial reporting information as well. When presenting above average or rising margins in financial statements, customers, suppliers, or labour unions could all potentially use this information to their advantage in their transactions and price negotiations with the disclosing firm. Since the usefulness of financial information diminishes over time, the proprietary costs also abate as time passes. Hence, if the penalty for late disclosure is smaller than the perceived proprietary cost, the firm has a clear incentive to delay disclosure.

Boiled down, these two theories as to why firms might want to delay the disclosure of their financial statements, despite having similar outcomes, originate from clearly distinguishable motivations. As described by Darrough and Stoughton [3], there are "conflicting objectives of (firms) with favorable and unfavorable information". Managers are likely inclined to obfuscate any agency problems, as this could raise the associated agency costs. Following the obfuscation theory and the related notion of 'bad news traveling slow', the catalyst for delaying disclosure is bad news. Therefore, we would expect a negative relation between filing lags and firm performance. The proprietary cost theory, in contrast, seems to be more relevant for firms performing better. Underperforming firms are less likely to be concerned about being the target of competitive accounting. Furthermore, weak margins are of little use to customers, suppliers, or trade unions in price negotiations. Therefore, following the proprietary cost theory, we could also expect a positive relation between filing lags and firm performance. The interaction between these two theories and their opposite relation to disclosure is captured in Figure 1 (adapted from Wittmann [9]).

*Hypotheses Development*

Prior studies provide us with theory and evidence on why companies delay the disclosure of financial information (see, e.g., Lukason et al. [8], Altman et al. [10], Whittred and Zimmer [11], Lawrence [12], Dye and Sridhar [19], Soltani [27], Owusu-Ansah and Leventis [28]). Findings for listed firms have shown that financial reporting delays are associated with financial distress. The focus of much of the research to date has been on

listed firms. In contrast, relatively few studies have been concerned with financial reporting delays in the context of private firms. Because private firms differ from listed firms in several respects, management motivations to disclose information may also be different. Since the demand for the financial information of private firms is lower [29,30], they may also have fewer incentives to disclose their financial statements in a timely manner. The recent evidence of Clatworthy and Peel [7] indeed found that the reporting behaviour of private UK firms is significantly influenced by the regulatory deadline. Similarly, Luypaert et al. [6] found that many private Belgian firms just meet the regulatory deadline and have few other incentives to file their financial statements on time.

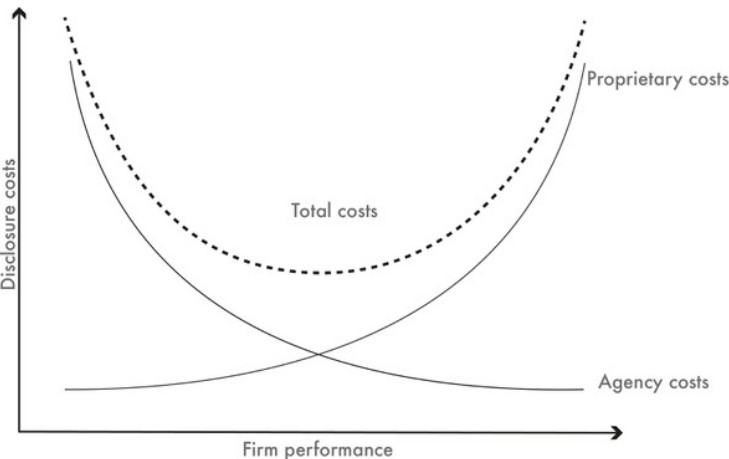

**Figure 1.** Firm performance and disclosure cost. Figure adapted from Wittmann [9].

The main reason why private firms delay the publication of financial statements is related to the idea that bad news travels slow. In the case of financial difficulties, firms may be inclined to delay the publication of unfavourable information. Altman et al. [10] studied the use of non-financial information for SMEs' credit risk models and found that the inclusion of the filing history of a firm significantly increases the predictive power of such models. An important finding was that late filings are a sign of SMEs entering financial distress, comparable to the findings of Lawrence [12] and Whittred and Zimmer [11]. The analysis showed that late filings are associated with a higher probability of bankruptcy [10]. The paper of Lukason et al. [8] examined the relation between late filing and financial distress among Estonian SMEs. Using the prominent model of Altman et al. [31] to predict financial distress, their findings suggested that firms which are more likely to experience financial distress have an increased chance of filing their financial reports after the regulatory deadline. The results of this study further showed that late filings are associated with lower liquidity and profitability ratios. Focusing exclusively on Belgian private firms' filing behaviour, Luypaert et al. [6] found evidence that supports the premise that bad news is disclosed later.

Building on the prior studies reviewed above, we propose the following hypothesis:

**Hypothesis 1.** *There is a negative association between financial health and late filing.*

The aforementioned previous literature has been mostly restricted to a relatively short time frame. In a private firm context, Luypaert et al. [6] as well as Clatworthy and Peel [7] found that past filing behaviour is the most important predictor of a firm's current filing behaviour. However, the time periods covered in their study encompass only two consecutive years. Taking into account the filing behaviour of firms over several years should provide a greater understanding of the motivations and incentives of filing delays. To date, however, no studies have been found which look into multiple filing lags over a longer time frame.

Whereas some firms may delay the publication of financial statements based on a deliberate ad hoc decision (e.g., because they want to delay the publication of unfavourable information), others may delay the publication of financial statements as a long-term strategy (e.g., to minimize proprietary costs of disclosure). More specifically, if the indirect costs of disclosing proprietary information outweigh the perceived costs of filing late, firms may choose to pursue a minimal disclosure strategy. In this case, firms may delay the publication of financial statements for multiple consecutive years, regardless of their financial health.

A recent study by Bernard et al. [5] showed that European private firms actively manage their firm size downward to avoid income statement disclosure. The authors concluded that "even small European firms appear to anticipate proprietary costs and incur substantial operational and financial costs to avoid them". This finding suggests that some managers of private firms follow a certain disclosure strategy where they weigh the costs and benefits of financial disclosure.

Based on this line of reasoning, we expect that the negative association between financial health and late filing becomes weaker when a firm delays the publication of financial statements year after year. In this case, we conjecture that the delay is more likely the result of a conscious disclosure strategy rather than an ad hoc decision based on the unfavourable information of a particular year. We therefore posit that multiple consecutive late filings moderates the relationship between financial health and late filing. Hence, we hypothesize the following:

**Hypothesis 2.** *The negative association between financial health and late filing is weaker for companies who file consistently late over the years.*

### 3. Research Design and Data

*3.1. Belgian Setting*

In accordance with the directive set out by the European Commission (DIRECTIVE 2003/58/EC), all limited liability firms in Belgium need to file their financial statements with the NBB. There are three types of financial statement formats for commercial firms, with increasing information requirements: the micro format, the abbreviated format, and the complete format. The appropriate format depends on criteria related to the size of the firm. To be eligible to use the micro format, firms cannot exceed more than one of following criteria: a staff headcount of less than 10 FTEs, a turnover of no more than 700,000 Euro, and a balance sheet total of no more than 350,000 Euro. Furthermore, to use the micro format, firms cannot have subsidiaries or be part of a group. The size criteria for the abbreviated format and the complete format are: a staff headcount of less than 50 FTEs, a turnover of no more than 9 million Euro, and a balance sheet total of no more than 4.5 million Euro. In order to be able to use the abbreviated format, firms cannot exceed more than one criterion. Exceeding more than one automatically results in the use of the complete format. No matter which format is used, all firms need to file their financial statements 1 month after the annual meeting of shareholders has approved it and no later than seven months after the closing of their accounting period. Despite the deadline of seven months, firms will only start to receive an administrative sanction for late filing when they file more than 8 months after the closing of their accounting period. These administrative sanctions gradually increase with the length of the filing lag and reach a maximum after more than twelve months after the closing of their accounting period. In total, there are two increments after the first administrative sanction deadline of 8 months: one after 9 months, and the other after twelve months after the closing of their accounting period. These administrative sanctions range from 120 to 1200 Euro, depending on the type of format used and the length of the filing lag. Next to these administrative sanctions, other possible consequences of late filing also include legal dissolution of the firm and penalties under civil law. If a firm files late, any damage suffered by third parties will be deemed to

arise from this misconduct, unless there is proof to the contrary. Furthermore, the burden of proof lies with the firm that failed to file on time.

### 3.2. Variables

We began by measuring the number of days between the filing of financial statements and the closing of the accounting period. To capture timeliness, we first constructed two dummy variables (LateDummy and ASDummy). Firms who file their financial statements more than seven months (around 213 days) after the closing of their accounting period were classified as late (LateDummy = 1). All other observations were classified as 'on-time' (LateDummy = 0). Financial statements that were filed after more than 8 months are classified as administrative sanctions (ASDummy = 1).

We constructed the variable SumLate to capture consistent late filings and firms' filing history. SumLate is a lagged variable which takes into account the eight previous years. We counted the total number of late filings for each individual firm starting from eight years before the year of investigation (t-8) until the year prior to investigation (t-1). As such, the maximum possible value of SumLate is 8, which would imply that a firm has filed late in all previous years.

Similarly, we also counted the total number of administrative sanctions (SumAS) for each individual firm starting from eight years before the year of investigation (t-8) until the year prior to investigation (t-1). Thus, our variable SumAS counts how many times in the 8 years prior to our year of investigation a firm filed their financial statements more than 8 months after the closing of their accounting period and, therefore, was subjected to an administrative sanction. Thus, similar to our SumLate variable, the maximum possible value of SumAS is 8.

From our lagged variables, SumLate and SumAS, we created two dummy variables, Late8 and AS8, respectively. These dummy variables took the value of 1 if a firm was late (Late8), or received an administrative sanction (AS8), every year in the previous 8 years. In all other cases they took the value of 0. In other words, Late8 (AS8) is equal to 1 when SumLate (SumAS) is equal to 8 in year 9 and 0 otherwise.

To measure firms' financial health, we employed the model of Altman et al. [31] We used the coefficients from model 2 from Altman et al. [31], a re-estimation of the model from Altman [32] using logistic regression analysis which is tailored to private firms. It is also used in the context of private firms' timeliness by Lukason et al. [8]. Specifically, our financial health score (Altman) was constructed out of four ratios: (X1) retained earnings/total assets, a measure for accumulated profitability; (X2) EBIT/total assets, expressing annual profitability; (X3) working capital/total assets, a measure for liquidity; and (X4) book value of equity/total debt, a measure for solvency. The score was calculated as $1/(1 + e^{-L})$, where $L = 0.035 - 0.862 \times X1 - 1.721 \times X2 - 0.495 \times X3 - 0.017 \times X4$. Furthermore, we controlled for firm size and age. Finally, there might be differences in the demand for (timely) financial information depending on the type of creditor. Hence, we also included the Trade.Debt (trade debt/total debt) and Fin.debt (financial debt/total debt) variables to control for the type of creditor. An overview of our employed variables is presented in Table 1.

### 3.3. Model Specification

We employed a cross-sectional logistic regression model for all our dependent variables. To control for outliers, all our continuous variables were winsorized at the 1% level. In order to create meaningful intercepts, continuous variables were centered. In all our models, we included the main effect of industry as a covariate. Analogous to Van den Bogaerd and Aerts [33], Ceustermans et al. [34], our industry classification has five industry dummies based on the two-digit NACE level.

**Table 1.** Variable definitions.

| Variables | Description |
| --- | --- |
| LateDummy | Dummy variable that is equal to 1 if the firm filed late in year 9 (more than seven months after closing their accounting period), and 0 otherwise. |
| ASDummy | Dummy variable that is equal to 1 if the firm was subjected to an administrative sanction in year 9 (filing after more than eight months after closing their accounting period), and 0 otherwise. |
| SumLate | Lagged sum of late filings (from the first observation (t-8) until year (t-1)). |
| SumAS | Lagged sum of administrative sanctions (from the first observation (t-8) until year (t-1)). |
| Late8 | Dummy variable equal to 1 for firms with 8 previous late filings and 0 otherwise. |
| AS8 | Dummy variable equal to 1 for firms with 8 previous administrative sanctions and 0 otherwise. |
| Altman | Altman score: Altman et al. [31] model 2. Note that a higher score should be interpreted as a lower financial health. |
| Size | Firm size measured as the natural logarithm of total assets. |
| Age | Firm age measured as the natural logarithm of the number of years since the date of incorporation. |
| Trade.Debt | The ratio of trade debt over total debt. |
| Fin.Debt | The ratio of financial debt over total debt. |
| Industry | Categorical variable that denotes the specific industry to which the firm belongs. |

To test our first hypothesis, we ran our cross-sectional logistic regression model without the interaction term (models 1 and 2). We expected that there would be a negative association between late filing and financial health. Hence, we would predict a positive coefficient for the Altman variable.

Model 1:

$$log\left(\frac{P(LateDummy_i = 1)}{1 - P(LateDummy_i = 1)}\right) = \begin{aligned}&\alpha_0 + \beta_1 Altman_i + \beta_2 Size_i + \beta_3 Age_i + \beta_4 Fin.Debt_i \\ &+ \beta_5 Trade.Debt_i + \beta_6 Industry_i + \varepsilon_i\end{aligned} \quad (1)$$

Model 2:

$$log\left(\frac{P(ASDummy_i = 1)}{1 - P(ASDummy_i = 1)}\right) = \begin{aligned}&\alpha_0 + \beta_1 Altman_i + \beta_2 Size_i + \beta_3 Age_i + \beta_4 Fin.Debt_i \\ &+ \beta_5 Trade.Debt_i + \beta_6 Industry_i + \varepsilon_i\end{aligned} \quad (2)$$

In order to test our second hypothesis, we were interested in testing the association between late filing and financial health as a function of the consistency of previous late filings. For this, we developed a model that gives nuanced insight into the effect of filing late in the previous eight years. That is, the variable SumLate is used as a consistency measure. By including this variable as a categorical predictor variable and inspecting its interaction with Altman, one can gain insights in the association between a firm's health and filing late in year 9 as a function of the specific number of times a firm was late in the previous 8 years. As we will show in the post hoc analyses, this model yields an excellent model quality (see Section 6 for a detailed description of this model and its results). We present our main model, in which the dummy variable Late8 is used instead of SumLate, in Section 5, allowing a more intuitive interpretation. Here the variable Late8 is 1 when a company filed late during all previous 8 years, and 0 otherwise. As we will show, similar results are found with this main model as when using the extended model (cfr. Section 6).

Consequently, we tested our second hypothesis by adding the interaction term Altman × Late8 and Altman × AS8 in model 3 and model 4, respectively. We expected that the negative association between financial health and late filing becomes weaker when a firm delays the publication of financial statements year after year. In other words, we expected that the negative association between financial health and being late is more prominent for companies who were not late consistently in the previous years. Conversely, for companies that were late every year in the previous 8 years, we expect that this might be explained by an underlying strategy rather than a firms' financial health. Furthermore,

due to the consequences associated with administrative sanctions, we expect this to be more pronounced for the interaction with the AS8 variable.

Model 3:

$$log\left(\frac{P(LateDummy_i = 1)}{1 - P(LateDummy_i = 1)}\right) = \begin{aligned}&\alpha_0 + \beta_1 Altman_i + \beta_2 Late8_i + \beta_3 Altman_i \times Late8_i + \beta_4 Size_i\\&+ \beta_5 Age_i + \beta_6 Fin.Debt_i + \beta_7 Trade.Debt_i + \beta_8 Industry_i + \varepsilon_i\end{aligned} \quad (3)$$

Model 4:

$$log\left(\frac{P(ASDummy_i = 1)}{1 - P(ASDummy_i = 1)}\right) = \begin{aligned}&\alpha_0 + \beta_1 Altman_i + \beta_2 AS8_i + \beta_3 Altman_i \times AS8_i + \beta_4 Size_i\\&+ \beta_5 Age_i + \beta_6 Fin.Debt_i + \beta_7 Trade.Debt_i + \beta_8 Industry_i + \varepsilon_i\end{aligned} \quad (4)$$

### 3.4. Data and Sample Breakdown

We obtained financial data between 2010 and 2020 for a large sample of Belgian private firms from a proprietary private database This extensive data set was collected by Companyweb, a company that specialises in collecting company information in Belgium. For the purposes of this study, Companyweb granted us confidential access to its data. The received data set contains all financial statement information of all limited liability companies in Belgium. We had a maximum of 9 observations for each firm. We excluded firms active in the financial sector due to other reporting requirements. This resulted in an initial database consisting of 3,390,051 firm-year observations. Cleaning the data for missing values resulted in a reduced sample size of 3,315,798 firm-year observations. We excluded firms that disclosed using the micro format, as these smaller firms are more likely to use an external accountant [35,36]. The use of an external accountant could possibly affect filing behaviour. Finally, we excluded firms with less than 9 observations. Only including firms with 9 observations allowed us to control for survivorship bias. Our final sample size consisted of 935,874 firm-year observations, which corresponds to 103,986 unique firms. An overview of our sample is presented in Table 2.

**Table 2.** Data Breakdown.

| Criteria | Drop | Sample |
|---|---|---|
| Belgian Private firms | | 3,390,051 |
| - Missing Values | −74,253 | |
| Sample | | 3,315,798 |
| - Firms disclosing in the micro format | −2,105,482 | |
| Sample | | 1,210,316 |
| - Data on nine consecutive years | −274,442 | |
| Final Sample | | 935,874 |
| Unique Firms | | 103,986 |

## 4. Results: Descriptive Statistics

Table 3 presents descriptive statistics for our dependent and independent variables. Table 4 reports Pearson's correlation coefficients for the variables. None of the coefficients are above the .600 mark, which is often used to assess potential multicollinearity problems [6]. Furthermore, it shows a positive correlation between the Altman score and our two dependent variables. This is a preliminary indication in support of our first hypothesis.

**Table 3.** Summary statistics of independent variables. (N = 103,986).

| Variable | Mean | Std. Dev. | P25 | P50 | P75 |
|---|---|---|---|---|---|
| LateDummy | 0.426 | | | | |
| ASDummy | 0.26 | | | | |
| Altman | 0.001 | 0.162 | −0.081 | −0.006 | 0.055 |
| Size | 0.008 | 1.748 | −0.951 | 0.023 | 1.087 |
| Age | 0.001 | 0.472 | −0.41 | 0.018 | 0.348 |
| Trade.Debt | 0.002 | 0.262 | −0.214 | −0.094 | 0.144 |
| Fin.Debt | 0 | 0.299 | −0.276 | −0.107 | 0.239 |
| SumLate | 2.838 | 2.765 | 0 | 2 | 5 |
| SumAS | 0.858 | 1.597 | 0 | 0 | 1 |
| Late8 | 0.089 | 0.284 | 0 | 0 | 0 |
| AS8 | 0.009 | 0.096 | 0 | 0 | 0 |

**Table 4.** Cross-correlation table. The variables are defined in Table 1. *p*-values are reported within brackets.

| | LateDummy | ASDummy | Altman | Size | Age | Trade.Debt | Fin.Debt | Late8 | AS8 |
|---|---|---|---|---|---|---|---|---|---|
| Altman | 0.057 | 0.046 | 1.000 | | | | | | |
| | (0.000) | (0.000) | | | | | | | |
| Size | −0.093 | −0.053 | −0.247 | 1.000 | | | | | |
| | (0.000) | (0.000) | (0.000) | | | | | | |
| Age | −0.035 | −0.013 | −0.040 | 0.223 | 1.000 | | | | |
| | (0.000) | (0.000) | (0.000) | (0.000) | | | | | |
| Trade.Debt | 0.016 | 0.017 | −0.149 | 0.017 | 0.072 | 1.000 | | | |
| | (0.000) | (0.000) | (0.000) | (0.000) | (0.000) | | | | |
| Fin.Debt | −0.002 | −0.007 | 0.126 | 0.203 | −0.038 | −0.393 | 1.000 | | |
| | (0.542) | (0.034) | (0.000) | (0.000) | (0.000) | (0.000) | | | |
| Late8 | 0.314 | 0.300 | 0.019 | −0.109 | −0.039 | 0.004 | −0.010 | 1.000 | |
| | (0.000) | (0.000) | (0.000) | (0.000) | (0.000) | (0.209) | (0.002) | | |
| AS8 | 0.105 | 0.140 | −0.007 | −0.022 | −0.001 | 0.008 | −0.007 | 0.310 | 1.000 |
| | (0.000) | (0.000) | (0.034) | (0.000) | (0.887) | (0.009) | (0.024) | (0.000) | |

*4.1. Late Filings*

Table 5 presents the breakdown of our SumLate variable. We can observe that 69% of all firms have filed late at least once in the 8 years prior to our year of analysis. In other words, only 31% of all firms have filed on time every year. The number of firms decreases with the number of late filings per firm, except for the category of persistent late filers (SumLate = 8). Interestingly, 9% of all firms, or 9223 unique firms, were late consistently every year, and this is higher compared to the number firms who file late 3 or more times in the eight-year period.

Next, we also compared the average financial health between firms with a different number of late filings within the eight-year period. A lower Altman score represents a better financial health situation. Table 5 shows that the financial health of firms deteriorates with the number of late filings they incur. However, firms who filed late consistently (8 out of 8 times) have a significantly lower Altman score on average (better financial health) compared to firms who file late 6 or 7 times. The average financial health of firms reaches a minimum around 6 late filings, after which the average financial health starts to increase again. This might be an early indication that not all late filings originate from motivations related to bad news.

**Table 5.** Table presenting the average Altman score of firms for each value of SumLate. The *p*-value represents the significance levels of the pairwise comparison using the Bonferroni method.

| SumLate | Firms | Percent | Cumulative | Altman | Delta | *p*-Value |
|---------|-------|---------|------------|--------|-------|-----------|
| 0 | 31,820 | 31% | 31% | 0.42868 | | |
| 1 | 14,320 | 14% | 44% | 0.44336 | 0.01468 | <0.001 |
| 2 | 10,448 | 10% | 54% | 0.45218 | 0.00882 | 0.001 |
| 3 | 9036 | 9% | 63% | 0.45577 | 0.00359 | 1.000 |
| 4 | 7852 | 8% | 71% | 0.46041 | 0.00464 | 1.000 |
| 5 | 7125 | 7% | 78% | 0.46504 | 0.00463 | 1.000 |
| 6 | 7116 | 7% | 84% | 0.46949 | 0.00445 | 1.000 |
| 7 | 7046 | 7% | 91% | 0.46771 | −0.00178 | 1.000 |
| 8 | 9223 | 9% | 100% | 0.45811 | −0.00960 | 0.006 |
| Total | 103,986 | 100% | | | | |

### 4.2. Administrative Sanctions

Similarly to Table 5, a breakdown of our SumAS variable is presented in Table 6. The focus on the administrative sanctions is interesting for a number of reasons. First, the consequences for the disclosing firm are more severe due to the monetary penalty associated with the administrative sanction. As a consequence, firms are likely to be more cautious with delaying the disclosure of their financial statements beyond this 8-month deadline. This makes it more likely that disclosing after 8 months is a deliberate strategy to withhold information and reduces the chance of it being a result of negligence. Furthermore, due to the fact that there is a clear cost related to disclosing after 8 months, firms are more likely to have consciously outweighed the benefits of late filing with the associated costs, especially when firms pay this 'price' for late filing year after year. This 'price' does not only refer to the monetary penalty, but also to the possible legal dissolution of the firm and the liability for any damage suffered by third parties as a result of the filing lag. Therefore, because we assume these late filings to be of a more deliberate nature, we expect a stronger relationship with our financial health variable.

**Table 6.** Table presenting the average Altman score of firms for each value of SumAS.The *p*-value represents the significance levels of the pairwise comparison using the Bonferroni method.

| SumAS | Firms | Percent | Cumulative | Altman | Delta | *p*-Value |
|-------|-------|---------|------------|--------|-------|-----------|
| 0 | 67,221 | 65% | 65% | 0.43862 | | |
| 1 | 16,537 | 16% | 81% | 0.46001 | 0.02138 | <0.001 |
| 2 | 7764 | 7% | 88% | 0.46922 | 0.00921 | 0.001 |
| 3 | 4372 | 4% | 92% | 0.47488 | 0.00566 | 1.000 |
| 4 | 2806 | 3% | 95% | 0.47984 | 0.00497 | 1.000 |
| 5 | 1919 | 2% | 97% | 0.47823 | −0.00161 | 1.000 |
| 6 | 1349 | 1% | 98% | 0.46936 | −0.00888 | 1.000 |
| 7 | 1053 | 1% | 99% | 0.46723 | −0.00212 | 1.000 |
| 8 | 965 | 1% | 100% | 0.43739 | −0.02984 | 0.001 |
| Total | 103,986 | 100% | | | | |

The results obtained from the preliminary analysis of the SumAS variable and the Altman score are summarised in Table 6. As we expected, compared to the SumLate variable there seems to be a stronger relationship with the financial health variable. The highest average Altman scores (lowest financial health) can be observed for firms with 4 administrative sanctions (compared to category 6 for SumLate). Furthermore, these values are higher compared to those presented in Table 5 (0.4798 for SumAS = 4, compared to 0.4695 for SumLate = 6). The pairwise comparison of the average Altman score shows that firms with 8 administrative sanctions have significantly lower Altman scores (i.e., are of

better financial health) compared to all other firms who received at least one administrative sanction. Although even those firms with 0 administrative sanctions have a higher Altman score (lower financial health), this difference was not found to be significant.

Taken together, these initial results from both our late filing variable and our administrative sanction variable would suggest that the relationship between late filings and financial health is dependent on the filing history of the firm. The finding, presented in Table 6, that firms who consistently pay a penalty for disclosing late (8 out of 8 times), are of better financial health on average compared to firms who paid a penalty only once, seems to contradict the obfuscation argument.

## 5. Results: Logistic Regression Analysis

The results of our baseline cross-sectional logistic regressions for the dependent variables LateDummy (model 1) and ASDummy (model 2) are presented in Tables 7 and 8, respectively. The results of our models where we included an interaction term between Altman and Late8 (model 3) and Altman and AS8 (model 4) are presented in Tables 9 and 10, respectively.

**Table 7.** Results of logistic regression with LateDummy (Model 1) as dependent variable. The variables are defined in Table 1. Continuous variables were centered. In order to preserve overview, coefficients for the industry variables are not reported. Sample corresp. Table 2. Corresp. significance levels: *** $p < 0.001$.

|  | B | SE | *p*-Value | 95% CI | exp(B) |
|---|---|---|---|---|---|
| Altman | 0.424 *** | 0.041 | <0.001 | [0.344; 0.505] | 1.529 |
| Size | −0.094 *** | 0.004 | <0.001 | [−0.101; −0.086] | 0.911 |
| Age | −0.082 *** | 0.014 | <0.001 | [−0.110; −0.055] | 0.921 |
| Trade.Debt | 0.198 *** | 0.028 | <0.001 | [0.144; 0.252] | 1.219 |
| Fin.Debt | 0.146 *** | 0.024 | <0.001 | [0.098; 0.193] | 1.157 |
| Industry | Added |  |  |  |  |
| Constant | −0.366 *** | 0.013 | <0.001 | [−0.392; −0.339] | 0.694 |
| Observations | 103,986 |  |  |  |  |
| Pseudo R-squared | 0.010 |  |  |  |  |
| Chi-squared | 1424.49 |  | <0.001 |  |  |
| AUC | 56.84% |  |  |  |  |

**Table 8.** Results of logistic regression with ASDummy (Model 2) as dependent variable. The variables are defined in Table 1. Continuous variables were centered. In order to preserve overview, coefficients for the industry variables are not reported. Sample corresp. Table 2. Corresp. significance levels: ** $p < 0.01$; *** $p < 0.001$.

|  | B | SE | *p*-Value | 95% CI | exp(B) |
|---|---|---|---|---|---|
| Altman | 0.489 *** | 0.045 | <0.001 | [0.400; 0.578] | 1.630 |
| Size | −0.053 *** | 0.004 | <0.001 | [−0.062; −0.044] | 0.948 |
| Age | −0.014 | 0.016 | 0.377 | [−0.045; 0.017] | 0.986 |
| Trade.Debt | 0.225 *** | 0.031 | <0.001 | [0.165; 0.286] | 1.253 |
| Fin.Debt | 0.072 ** | 0.027 | 0.008 | [0.018; 0.125] | 1.074 |
| Industry | Added |  |  |  |  |
| Constant | −1.166 *** | 0.015 | <0.001 | [−1.196; −1.136] | 0.312 |
| Observations | 103,986 |  |  |  |  |
| Pseudo R-squared | 0.005 |  |  |  |  |
| Chi-squared | 630.4 |  | <0.001 |  |  |
| AUC | 55.13% |  |  |  |  |

The following interpretations are made, assuming that all other variables are held constant. We start with the results of our baseline models in Tables 7 and 8. The parameter

estimates for model 1 can be found in Table 7. Recall that the continuous variables were centered and the reference category for industry is 1. The intercept from model 1 is estimated to be −0.366 (B = −0.366, *p* = 0.013, exp(B) = 0.694), indicating that the odds of being late in year nine for a company with an average health (Altman = 0) and average values for our control variables (Age, Size, Fin.Debt, Trade.Debt) from industry 1 is 0.694. The positive Altman estimate in Model 1 indicates that lower levels of financial health are associated with higher odds of being late (B = 0.424, *p* < 0.001, exp(B) = 1.529). Likewise, from Table 8 it is clear that the odds of filing after the administrative sanction deadline are higher for firms that have a lower financial health (cfr. Altman Model 2; B = 0.489, *p* < 0.001, exp(B) = 1.630). These results are consistent with other research, which found a negative relation between a firm's financial health and late filing [6,8,10,12].

Regarding our control variables, the negative coefficient for Size in both model 1 (B = −0.094, *p* < 0.001, exp(B) = 0.911) and model 2 (B = −0.053, *p* < 0.001, exp(B) = 0.948) suggest that the odds of filing late, and the odds of receiving an administrative sanction, are lower for smaller firms. The negative coefficient for Size in both baseline models is in line with earlier results [6,9]. The negative coefficient for Age in model 1 (B = −0.082, *p* < 0.001, exp(B) = 0.921) implies that the odds of filing late are lower for older firms. This result matches those observed in previous studies (e.g., [6–8]). Note that the parameter estimate of Age in model 2 did not differ significantly from zero (B = −0.014, *p* = 0.377, exp(B) = 0.986). Therefore, no significant effect from Age was found on the odds of receiving an administrative sanction. The positive coefficients for both Trade.Debt and Fin.Debt. match those observed in earlier studies [6,28]. This indicates that for companies with more trade debt or financial debt, the odds of filing late or receiving administrative sanctions are higher .

**Table 9.** Results of logistic regression with LateDummy (Model 3) as dependent variable. The variables are defined in Table 1. Continuous variables were centered. In order to preserve overview, coefficients for the industry variables are not reported. Sample corresp. Table 2. Corresp. significance levels: *** *p* < 0.001.

| | **B** | **SE** | ***p*-Value** | **95% CI** | **exp(B)** |
|---|---|---|---|---|---|
| Altman | 0.562 *** | 0.044 | <0.001 | [0.475; 0.648] | 1.754 |
| Late8 | 2.955 *** | 0.040 | <0.001 | [2.876; 3.034] | 19.202 |
| Altman×Late8 | −1.132 *** | 0.225 | <0.001 | [−1.573; −0.692] | 0.322 |
| Size | −0.060 *** | 0.004 | <0.001 | [−0.068; −0.051] | 0.942 |
| Age | −0.066 *** | 0.015 | <0.001 | [−0.095; −0.038] | 0.936 |
| Trade.Debt | 0.196 *** | 0.029 | <0.001 | [0.139; 0.253] | 1.216 |
| Fin.Debt | 0.116 *** | 0.025 | <0.001 | [0.066; 0.166] | 1.123 |
| Industry | Added | | | | |
| Constant | −0.561 *** | 0.014 | <0.001 | [−0.589; −0.533] | 0.571 |
| Observations | 103,986 | | | | |
| Pseudo R-squared | 0.086 | | | | |
| Chi-squared | 12,137.72 | | <0.001 | | |
| AUC | 63.62% | | | | |
| LR-test | 10,713.24 | | <0.001 | | |

**Table 10.** Results of logistic regression with ASDummy (Model 4) as dependent variable. The variables are defined in Table 1. Continuous variables were centered. In order to preserve overview, coefficients for the industry variables are not reported. Sample corresp. Table 2. Corresp. significance levels: * $p < 0.05$; ** $p < 0.01$; *** $p < 0.001$.

|  | **B** | **SE** | ***p*-Value** | **95% CI** | **exp(B)** |
|---|---|---|---|---|---|
| Altman | 0.531 *** | 0.046 | <0.001 | [0.441; 0.621] | 1.701 |
| AS8 | 3.189 *** | 0.105 | <0.001 | [2.982; 3.395] | 24.255 |
| Altman×AS8 | −1.799 ** | 0.588 | 0.002 | [−2.952; −0.647] | 0.165 |
| Size | −0.049 *** | 0.005 | <0.001 | [−0.058; −0.040] | 0.952 |
| Age | −0.018 | 0.016 | 0.254 | [−0.049; 0.013] | 0.982 |
| Trade.Debt | 0.217 *** | 0.031 | <0.001 | [0.157; 0.278] | 1.243 |
| Fin.Debt | 0.068 * | 0.027 | 0.013 | [0.015; 0.122] | 1.071 |
| Industry | Added |  |  |  |  |
| Constant | −1.194 *** | 0.016 | <0.001 | [−1.224; −1.163] | 0.303 |
| Observations | 103,986 |  |  |  |  |
| Pseudo R-squared | 0.020 |  |  |  |  |
| Chi-squared | 2367.97 |  | <0.001 |  |  |
| AUC | 56.47% |  |  |  |  |
| LR-test | 1737.57 |  | <0.001 |  |  |

Tables 9 and 10 present the results of our models with the interaction term to test our second hypothesis. The reported output from the likelihood ratio tests confirms that the inclusion of the interaction term significantly improves our models compared to the models without the interaction term (LR test statistics of 10,713.24 and 1737.57 respectively. *p*-values < 0.001). With respect to the control variables, these are largely in line with those reported in our baseline models (model 1 and model 2).

Regarding our financial health variable in model 3 (Table 9), the positive Altman coefficient (B = 0.562, $p < 0.001$, exp(B) = 1.754) indicates that, for firms who did not file late every year in the previous 8 years (Late8 = 0), higher Altman scores are associated with higher odds of filing late. That is, the lower the financial health of a company that has not always filed late in the 8 previous years, the higher the odds that they will be late in year 9. This is in line with the results reported in our baseline models (Tables 7 and 8).

The positive parameter estimate for our dummy variable Late8 indicates that we will have a higher intercept for that group of firms (−0.561 + 2.955 = 2.394). Thus, the odds of being late in year nine for a company that has been late consistently in the eight previous years (Late8 = 1) with an average health (Altman = 0), and average values for our control variables (Age, Size, Fin.Debt, Trade.Debt) from industry 1 are higher compared to firms who were not always late in the past (Late8 = 0). The odds of filing late in year 9 are estimated to be almost 20 times larger for companies that were consistently late before as compared to companies that were not always late. Note that this difference was found to be significant (B = 2.955, $p < 0.001$, exp(B) = 19.202). The same interpretation applies to the parameter estimate for our AS8 variable from model 4 (B = 3.189, $p < 0.001$, exp(B) = 24.255).

Regarding our interaction term from model 3, we have another estimate for Altman for the group of firms that has always filed late in the past (Late8 = 1). For a company that was always late in the previous 8 years, the parameter estimate for Altman is now negative (0.562 − 1.132 = −0.570) and this estimate differs significantly from the one for the category of companies that was not always late (B = −1.132, $p < 0.001$, exp(B) = 0.322). In other words, we find an opposite association between financial health and late filing for the group of firms that filed late consistently in the past (Late8 = 1). The negative estimate indicates that, for companies that were always late before, lower Altman scores are associated with higher odds of being late. That is, the healthier a company is that has always been late before, the higher the odds that they will be late in the subsequent year,

all other variables held constant. This positive association between financial health and late filing is opposite to the ones we find in our baseline models.

The results from model 4 (Table 10), which focuses on administrative sanctions instead of late filings, are consistent with the results of model 3. The positive Altman coefficient indicates that, for firms who did not receive an administrative sanction consistently in the previous eight years (AS8 = 0), higher Altman scores are associated with higher odds of filing after the administrative sanction deadline (B = 0.531, $p < 0.001$, exp(B) = 1.701). The opposite is true for firms who received an administrative sanction every year in the past (AS8 = 1). The parameter estimate for firms with eight previous administrative sanctions is negative (0.531 − 1.799 = −1.268). That is, the healthier a company is that always received an administrative sanction in the 8 previous years, the higher the odds that they will be late in year 9.

In general, from the previous description, it should be clear that the parameter estimates of our dummy variables (Late8 and AS8) show the difference in intercept between the group of firms with less than eight previous late filings (administrative sanctions) and the reference category (Late8 = 1 and AS8 = 1). The interaction terms between Altman and Late8 (AS8) show the difference in the association between Altman and being late (filing after the administrative sanction deadline) in year 9 between the group of firms with less than 8 previous late filings (administrative sanctions) and the reference category.

Taken together, the results presented in Table 9 confirm our second hypothesis and show that the negative association between the odds of late filing and financial health is weaker for companies who file consistently late over the years. More than that, the results suggest that the association firms' health and filing late (in year 9) is opposite (i.e., positive) for firms that file late consistently. In the models presented in Section 6, we add the interaction term Altman×SumLate and Altman×SumAS in model 3b and model 4b, respectively. The results of these extended models are presented in Tables 11 and 12 and are also graphically depicted in Figure 2. As Models 3b and 4b reveal, we obtain similar findings when the dummy variables Late8 and AS8 are measured as categorical variables instead. Importantly, the coefficients of Altman and both the interaction terms with SumLate and SumAS are highly significant across all reported models. For our variables SumLate and SumAS we selected 8 (i.e., firms that filed consistently late and consistently received an administrative sanction, respectively) as the reference category in our models 3b and 4b. Note that this differs from models 3 and 4 in which 0 (i.e., firms that did not filed consistently late and firms that did not receive an administrative sanction, respectively) was chosen as the reference category. Therefore, the signs for Altman and our consistency measures (SumLate and SumAS) are opposite as compared to models 3 and 4.

**Table 11.** Results of logistic regression with LateDummy (Model 3b) as dependent variable. The variables are defined in Table 1. Continuous variables were centered. For the lag-variables, the reference category is the companies that were always late in the past 8 years (SumLate = 8). In order to preserve overview, coefficients for the industry variables are not reported. 95% confidence intervals are based on the likelihood ratio test. Sample corresp. Table 2. Corresp. significance levels: * $p < 0.05$; ** $p < 0.01$; *** $p < 0.001$.

|  | B | SE | $p$-Value | 95% CI | exp(B) |
|---|---|---|---|---|---|
| Altman | −0.338 | 0.216 | 0.118 | [−0.758; 0.091] | 0.713 |
| SumLate = 0 | −4.322 *** | 0.043 | <0.001 | [−4.407; −4.239] | 0.013 |
| SumLate = 1 | −3.477 *** | 0.044 | <0.001 | [−3.563; −3.392] | 0.031 |
| SumLate = 2 | −2.995 *** | 0.044 | <0.001 | [−3.082; −2.908] | 0.050 |
| SumLate = 3 | −2.685 *** | 0.045 | <0.001 | [−2.774; −2.598] | 0.068 |
| SumLate = 4 | −2.344 *** | 0.046 | <0.001 | [−2.434; −2.256] | 0.096 |
| SumLate = 5 | −1.999 *** | 0.047 | <0.001 | [−2.091; −1.909] | 0.135 |
| SumLate = 6 | −1.615 *** | 0.047 | <0.001 | [−1.709; −1.523] | 0.199 |
| SumLate = 7 | −1.105 *** | 0.050 | <0.001 | [−1.203; −1.008] | 0.331 |
| Altman × SumLate = 0 | 0.745 ** | 0.241 | 0.002 | [0.269; 1.214] | 2.107 |

**Table 11.** *Cont.*

|  | **B** | **SE** | ***p*-Value** | **95% CI** | **exp(B)** |
|---|---|---|---|---|---|
| Altman × SumLate = 1 | 1.050 *** | 0.247 | <0.001 | [0.563; 1.530] | 2.858 |
| Altman × SumLate = 2 | 0.774 ** | 0.249 | 0.002 | [0.283; 1.258] | 2.168 |
| Altman × SumLate = 3 | 0.684 ** | 0.251 | 0.007 | [0.188; 1.173] | 1.981 |
| Altman × SumLate = 4 | 0.556 * | 0.257 | 0.030 | [0.050; 1.057] | 1.744 |
| Altman × SumLate = 5 | 0.290 | 0.259 | 0.263 | [−0.221; 0.795] | 1.336 |
| Altman × SumLate = 6 | 0.383 | 0.264 | 0.147 | [−0.138; 0.899] | 1.467 |
| Altman × SumLate = 7 | 0.180 | 0.275 | 0.512 | [−0.361; 0.718] | 1.198 |
| Size | 0.008 | 0.005 | 0.097 | [0.064; 0.129] | 1.008 |
| Age | 0.096 *** | 0.016 | <0.001 | [−0.001; 0.017] | 1.101 |
| Trade.Debt | 0.158 *** | 0.032 | <0.001 | [−0.061; 0.050] | 1.171 |
| Fin.Debt | −0.005 | 0.028 | 0.850 | [0.094; 0.221] | 0.995 |
| Industry | Added |  |  |  |  |
| Constant | 2.435 *** | 0.042 | <0.001 | [2.354; 2.518] | 11.416 |
| Observations | 103,986 |  |  |  |  |
| Pseudo R-squared | 0.227 |  |  |  |  |
| AUC | 80.44% |  |  |  |  |
| LR-test | 38.37 | (<0.001) |  |  |  |

**Table 12.** Results of logistic regression with ASDummy (Model 4b) as dependent variable. The variables are defined in Table 1. Continuous variables were centered. For the lag-variables, the reference category are the companies that received an administrative sanction every year in the past (SumAS = 8). In order to preserve overview, coefficients for the industry variables are not reported. 95% confidence intervals are based on the likelihood ratio test. Sample corresp. Table 2. Corresp. significance levels: * $p < 0.05$; ** $p < 0.01$; *** $p < 0.001$.

|  | **B** | **SE** | ***p*-Value** | **95% CI** | **exp(B)** |
|---|---|---|---|---|---|
| Altman | −1.212 * | 0.580 | 0.037 | [−2.327; −0.047] | 0.298 |
| SumAS = 0 | −3.806 *** | 0.105 | <0.001 | [−4.019; −3.605] | 0.022 |
| SumAS = 1 | −2.835 *** | 0.106 | <0.001 | [−3.049; −2.632] | 0.059 |
| SumAS = 2 | −2.387 *** | 0.107 | <0.001 | [−2.604; −2.182] | 0.092 |
| SumAS = 3 | −2.018 *** | 0.109 | <0.001 | [−2.238; −1.809] | 0.133 |
| SumAS = 4 | −1.851 *** | 0.112 | <0.001 | [−2.076; −1.637] | 0.157 |
| SumAS = 5 | −1.463 *** | 0.116 | <0.001 | [−1.695; −1.240] | 0.232 |
| SumAS = 6 | −1.287 *** | 0.121 | <0.001 | [−1.528; −1.054] | 0.276 |
| SumAS = 7 | −0.839 *** | 0.129 | <0.001 | [−1.095; −0.588] | 0.432 |
| Altman × SumAS = 0 | 1.779 ** | 0.584 | 0.002 | [0.608; 2.902] | 5.925 |
| Altman × SumAS = 1 | 1.381 * | 0.588 | 0.019 | [0.201; 2.513] | 3.978 |
| Altman × SumAS = 2 | 1.078 | 0.594 | 0.069 | [−0.112; 2.221] | 2.940 |
| Altman × SumAS = 3 | 1.034 | 0.605 | 0.087 | [−0.176; 2.199] | 2.812 |
| Altman × SumAS = 4 | 0.817 | 0.614 | 0.183 | [−0.411; 2.001] | 2.264 |
| Altman × SumAS = 5 | 1.173 | 0.637 | 0.065 | [−0.097; 2.403] | 3.231 |
| Altman × SumAS = 6 | 0.815 | 0.665 | 0.221 | [−0.509; 2.103] | 2.259 |

**Table 12.** *Cont.*

|  | **B** | **SE** | *p*-**Value** | **95% CI** | **exp(B)** |
|---|---|---|---|---|---|
| Altman × SumAS = 7 | 1.086 | 0.698 | 0.120 | [−0.298; 2.441] | 2.962 |
| Size | −0.022 *** | 0.005 | <0.001 | [0.024; 0.091] | 0.978 |
| Age | 0.057 ** | 0.017 | 0.001 | [−0.032; −0.013] | 1.059 |
| Trade.Debt | 0.104 ** | 0.033 | 0.002 | [−0.013; 0.101] | 1.109 |
| Fin.Debt | 0.044 | 0.029 | 0.130 | [0.038; 0.169] | 1.045 |
| Industry | Added |  |  |  |  |
| Constant | 2.041 *** | 0.106 | <0.001 | [1.838; 2.254] | 7.695 |
| Observations | 103,986 |  |  |  |  |
| Pseudo R-squared | 0.120 |  |  |  |  |
| Chi-squared | 14,001.24 | (<0.001) |  |  |  |
| AUC | 71.24% |  |  |  |  |
| LR-test | 59.84 | (<0.001) |  |  |  |

## 6. Post Hoc Analyses

In this section, we developed a model that gives a more nuanced insight into the effect of filing late in the previous eight years by using the variables SumLate and SumAS as a measure for consistency. In particular, a more nuanced association between firms' health (Altman) and the odds of filing late in year 9 can be discovered when taking into account the exact number of times a firm filed late before. For this reason, SumLate and SumAS were treated as categorical variables in the following analyses. More specifically, to test our second hypothesis, we added the interaction term Altman*SumLate and Altman*SumAS in model 3b and model 4b, respectively. By including these variables as categorical predictor variables and inspecting their interaction with Altman, one can gain insights in the association between a firm's health and filing late in year 9 as a function of the specific number of times a firm was late in the previous 8 years. We expected that the negative association between financial health and late filing becomes weaker when a firm delays the publication of financial statements year after year. In other words, we expected that the association between financial health and being late is more prominent for companies with lower values of being late in the previous years.

Regarding the reference group of our categorical variables SumLate and SumAS, we needed to consider our second hypothesis. We were particularly interested in comparing the consistent late filers (i.e., SumLate and SumAS equal to 8) with firms who were not consistently late, in order to observe whether the relationship with financial health is significantly different from other late filers (i.e., other values of SumLate and SumAS). Therefore, for our variables SumLate and SumAS we selected 8 as the reference category (8 late filings and 8 administrative sanctions, respectively). Note that the selection of the reference group does not change anything regarding the significance of the main and interaction terms.

Model 3b:

$$log\left(\frac{P(LateDummy_i = 1)}{1-P(LateDummy_i = 1)}\right) = \begin{aligned}&\alpha_0 + \beta_1 Altman_i + \beta_2 SumLate_i + \beta_3 Altman_i \times SumLate_i + \beta_4 Size_i \\ &+ \beta_5 Age_i + \beta_6 Fin.Debt_i + \beta_7 Trade.Debt_i + \beta_8 Industry_i + \varepsilon_i\end{aligned} \tag{5}$$

Model 4b:

$$log\left(\frac{P(ASDummy_i = 1)}{1-P(ASDummy_i = 1)}\right) = \begin{aligned}&\alpha_0 + \beta_1 Altman_i + \beta_2 SumAS_i + \beta_3 Altman_i \times SumAS_i + \beta_4 Size_i \\ &+ \beta_5 Age_i + \beta_6 Fin.Debt_i + \beta_7 Trade.Debt_i + \beta_8 Industry_i + \varepsilon_i\end{aligned} \tag{6}$$

The Type II analysis of variance tests (untabulated) for Model 3b show a significant main effect of financial health ($\chi_1^2 = 32.8$, $p < 0.001$), SumLate ($\chi_8^2 = 30818.3$, $p < 0.001$), the interaction of Altman×SumLate ($\chi_8^2 = 38.4$, $p < 0.001$), age ($\chi_1^2 = 34.6$, $p < 0.001$), trade.debt

($\chi_1^2 = 23.6$, $p < 0.001$) and industry ($\chi_4^2 = 57.7$, $p < 0.001$). No significant effect of size ($\chi_1^2 = 2.8$, $p = 0.097$) nor fin.debt ($\chi_1^2 = 0.0$, $p = 0.850$) was found.

The Type II analysis of variance tests (untabulated) for Model 4b show a significant main effect of financial health ($\chi_1^2 = 21.5$, $p < 0.001$), SumAS ($\chi_8^2 = 13311.0$, $p < 0.001$), the interaction of Altman×SumAS ($\chi_8^2 = 59.8$, $p < 0.001$), age ($\chi_1^2 = 11.6$, $p < 0.001$), size ($\chi_1^2 = 21.7$, $p < 0.001$), trade.debt ($\chi_1^2 = 9.6$, $p = 0.002$) and industry ($\chi_4^2 = 74.9$, $p < 0.001$). No significant effect of fin.debt ($\chi_1^2 = 2.3$, $p = 0.130$) was found.

Tables 11 and 12 present the results of our models with the categorical interaction terms to test our second hypothesis. The results from the interaction with SumLate, presented in Table 11, are graphically depicted in Figure 2 as well. With respect to the control variables, the results are largely in line with those reported in our baseline models (model 1 and model 2). Regarding our financial health variable in model 3b (Table 11), financial health was a significant predictor for the odds of being late in year 9 ($\chi_1^2 = 32.8$, $p < 0.001$). More detailed insights can be gained when inspecting the parameter estimates. These estimates show that, in the category of companies that were always late in the previous years (cfr.SumLate = 8 is the reference category), no significant association between Altman and the odds of being late can be found in year 9 (B = −0.338, $p = 0.118$, exp(B) = 0.713).

Stated differently, for the group of companies that were always late in the previous 8 years, no effect of the financial health of the company was found. This provides evidence for the hypothesis that in this category, the odds of being late is not associated with lower financial health.

When another category of SumLate is of interest, we have another intercept and another estimate for Altman. For example, for a company that was never late in the previous 8 years (SumLate = 0), the parameter estimate for Altman is positive (−0.338 + 0.745 = 0.407). The positive estimate indicates that, for companies that were never late before, higher scores of Altman are associated with higher odds of being late. That is, the unhealthier a company is that was never late before, the higher the odds that they will be late in year 9. One may also note that the effect of financial health on the odds of being late in year 9 differs significant between the companies that were never late and those that were always late before (B = 0.745, $p < 0.001$). We can also observe that there is no significant difference between the parameter estimates of Altman for those firms who were always late (SumLate = 8) and those who were late 5, 6, or 7 times (all $p$-values > 0.05).

The results from model 4b (Table 12), which focuses on administrative sanctions instead of late filings, are consistent with the results of model 3b. The negative Altman coefficient ($p = 0.037$) indicates that, for firms who received an administrative sanction every year in the previous 8 years (SumAS = 8), lower scores of Altman are associated with higher odds of filing after the administrative sanction deadline. That is, the healthier a company is that has always received an administrative sanction in the 8 previous years, the higher the odds that they will be late in year 9. The opposite is true for firms who never received an administrative sanction (SumAS = 0). The parameter estimate for firms with 0 previous administrative sanctions is positive (−1.212 + 1.779 = 0.567). It should be noted, however, that the models with SumAS as the categorical variable (model 2 and model 4) have lower explanatory power.

In general, from the previous description it should be clear that the parameter estimates of our categorical variables (SumLate and SumAS) show the difference in intercept between that category of the concerning categorical variable and the reference category (SumLate = 8 and SumAS = 8). The interaction terms between Altman and SumLate (SumAS) show the difference in the association between Altman and being late (filing after the administrative sanction deadline) in year 9 for that category of SumLate (SumAS) and the reference category. Using this information, similar interpretations can be made by the interested reader for the other categories of SumLate. The results are also graphically depicted in Figure 2, from which it should be clear that the effect of financial health on the odds of filing late in year 9 is different for firms with different filing histories. Figure 2 shows that the probability to file late increases with the number of times a firm filed late in

the previous 8 years. This represents the different intercepts for each category of SumLate. The slopes of each individual category represent the parameter estimates of Altman for that concerning category of SumLate. For the groups of firms with 0, 1, 2, 3, or 4 late filings in the previous 8 years, it can be observed that higher values of Altman are associated with a higher probability to file late (i.e., a positive slope). For the groups of firms with 5, 6, 7, or 8 late filings, the plots seem practically horizontal. This illustrates that there is no significant effect of financial health on the odds of filing late for those groups of firms. Hence, firms who were late 5, 6, 7 or 8 times do not seem to file late because of lower financial health issues.

Taken together, the results presented in Table 11 confirm our second hypothesis and show that the relationship between late filing and financial health is weaker for companies who file consistently late over the years. More than that, the results suggest that firms' health is irrelevant with respect to filing late (in year 9), not only for the firms that file late consistently, but for firms with more than 4 late filings in the past 8 years as well.

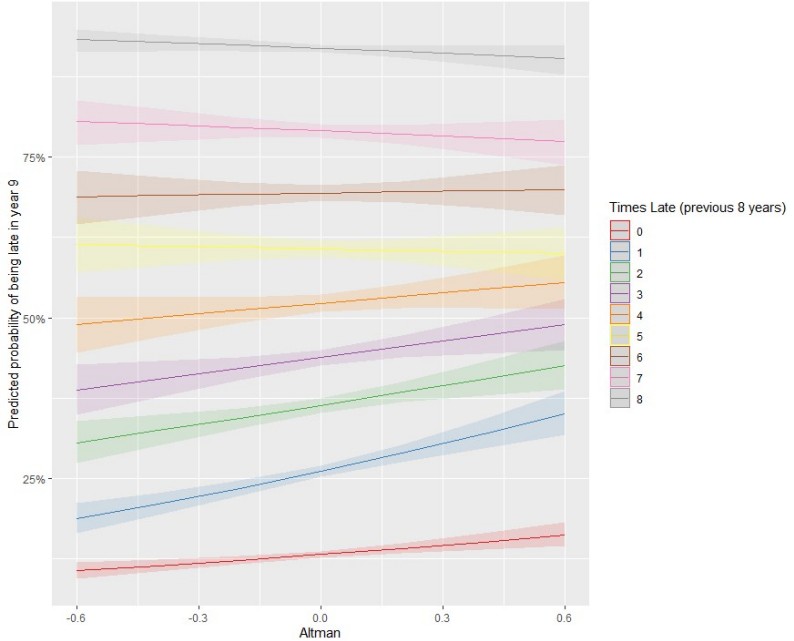

**Figure 2.** Graph showing the probability of filing late in year 9 for each category of SumLate in relation to the Altman score. Confidence interval shown as shaded region.

## 7. Discussion

This work contributes in several ways to the existing knowledge of timeliness of financial reporting by providing the first comprehensive investigation of late filing behaviour over a longer time frame. Prior research has recognized the impact of past filing behaviour on current filing behaviour, but the time periods covered in previous studies encompass, at most, two consecutive years. To the best of our knowledge, no study investigated multiple late filings over a longer time frame.

First of all, our data reveals that consistent late filings are not uncommon. Over 8 years, 9223 firms (9% of our sample) have filed late every single year. Our results not only confirm previous studies' findings that late filing is associated with lower financial health, but we also find that the relationship between late filing and financial health is opposite for companies who file consistently late over the years. For those group of firms, lower financial health is associated with a lower probability of filing late (after the administrative sanction deadline).

The opposite association that we find for the groups of consecutive late filers could be attributed to a different nature of the underlying motivation to disclose late. Luypaert et al. [6] find that unfavorable information is disclosed later and thus argue that firms realize that the information contained in financial statements is being used by stakeholders.

However, following the arguments from the proprietary cost theory and those made by the literature on competitive accounting, the same reasoning could also apply to delaying the disclosure of neutral or favorable information. The difference with delaying the disclosure of unfavorable information stems from the fact that it is not necessarily a choice that is made on a year-to-year basis, depending on the firm performance of a specific year. Instead, these kind of motivations to disclose could be part of a long-term corporate strategy. This could explain the difference we find between consistent/consecutive late filings and infrequent/sporadic late filings.

Prior studies have shown that past filing behaviour is one of the most important predictors of current filing behaviour [6]. We confirm this finding. Furthermore, our results provide support for the conceptual premise that there might be different motivations to disclose late. The idea that firms' disclosing behaviour in itself contains relevant intrinsic information about the firm is not new. Minnis and Shroff [4] argue that firms' disclosure practises and decisions could carry information about that firm. This study is build upon and further explores this idea. In sum, our findings add to the literature by providing a new understanding of how past filing behaviour, when examined over a longer period of time, could aid in uncovering the motivations behind late filings. These different motivations are associated with distinctive firm characteristics.

Our results corroborate previous investigations on timeliness which have found no evidence that highly leveraged companies are more likely to report on-time [6,8,28]. These results support the premise that private firms are more likely to use other (i.e., private) information channels to resolve information asymmetries with their creditors [29,30]. The argument that the usefulness of financial statements as a tool to predict future performance and cash flows is deteriorating [37,38] might also help to explain our results, as this is one of the main uses of financial statements for creditors [39] and consequently would make timeliness a less important issue.

Several questions still remain to be answered. A natural progression of this work is to analyse if competition affects filing behaviour. A qualitative study could examine if delaying disclosure is a technique used by managers to actively reduce the informational value of financial statements to stakeholders, and furthermore, if this technique is associated with consistent late filings over a longer time frame. Doing so would allow the distinction between infrequent late filings, which we expect is more likely to arise out of managers' inclination to obfuscate bad performance, and consistent late filings to be substantiated. Furthermore, a greater focus on the corporate governance might produce interesting findings that account more for the effect of the management on disclosure practises. Previous studies have already indicated that corporate governance affects firm performance, agency problems, and risk [40,41], and thus, potentially filing practises as well. Moreover, Giroud and Mueller [42] find that the effects of corporate governance for a firm are related to the level of competition in the industry. Taking into account the interplay between competition and corporate governance is a fruitful area for future research. Finally, incorporating data on the use of information technology (IT) could further improve the validity of our results. Previous literature has showed that the use of IT affects the timeliness of financial reporting by improving efficiency [43–45].

## 8. Conclusions

With this paper we provided further evidence on the relationship between late filing and financial health. First, we confirmed the negative association between firms' health and late filing. More importantly, this is the first study to show that this negative association does not apply to firms who are late consistently. This has not previously been described. Our results suggest a positive association between firms' health and filing late for the group of firms who filed late consistently in the past (8 years before). Consequently, these findings challenge the idea that late filing is automatically associated with delaying unfavorable information.

Previous literature seems to focus on the idea that disclosing late is a result of management trying to obfuscate bad performance. The possibility that there exist other motives to disclose late is largely neglected. Instead our results show that, apart from obfuscating bad performance, there seem to be different motivations that incentivise firms to disclose their financial statements after the legal deadline. These other motivations could perhaps be better explained by the proprietary cost theory. However, our model is based on publicly available information and does not allow this theory to be confirmed. Extending our analysis with a measure for competition, to test for any potential proprietary costs, could therefore be an interesting focus for future research.

Finally, these findings have important implications. For example, Altman et al. [10] showed that reporting lags are associated with financial distress and that including this type of information benefits the predictive power of credit risk models. Our findings would suggest that taking into account the history of firms' financial reporting behaviour (i.e., how many times they have filed late before) could significantly alter the interpretation of such information and therefore potentially further improve credit risk models.

**Author Contributions:** Conceptualization, T.S. and S.C.; Data curation, T.S. and L.S.; Formal analysis, T.S. and L.S.; Methodology, T.S. and S.C.; Supervision, S.C.; Visualization, T.S. and L.S.; Writing—original draft preparation, T.S. and S.C.; Writing—review and editing, S.C. All authors have read and agreed to the published version of the manuscript.

**Funding:** This research received no external funding.

**Data Availability Statement:** The data that support the findings of this study are available from the corresponding author upon reasonable request.

**Conflicts of Interest:** The authors declare no conflict of interest.

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
