# Peer review of "Not All Late Filers Are the Same: Distinguishing between Differences in Filing Behaviour"

_sustainability, doi:10.3390/su131910862_

Round 1

Reviewer 1 Report

See attached review report, as well as annotated version of the manuscript.

Reviewer 2 Report

Dear Authors,

The article has a bad structure and already at the first reading stage. The introduction contains things that should be included in literature review and in the discussion and conclusions section (e.g. "Our findings suggest"). The authors did not take care of the correct formatting of the text, which was scattered in many places. For instance lines 15-16, Figure 1.

Moreover, Figure 2 is unreadable. There are no sources in the figures and tables.

The number of references is very small. There are no DOI numbers. Moreover, as many as 10 out of 23 are literature items from before 2000.

Reviewer 3 Report

attached

Round 2

Reviewer 1 Report

This is a revision (R.1) of a paper submitted to Sustainability. While the authors tried to incorporate the input I provided in the previous round, I am not happy with (some of) the changes made. The authors keep estimating models in inappropriate ways. They have, indeed, aligned part of the research design with the suggestions made in the previous review round. However, none of the reported models is estimated in a proper manner. For the main models, the dependent and independent variables keep being flawed (i.e., the meaning of the dependent is fundamentally different for year 1 (max = 1) compared to year 9 (max = 9), in a similar way, the meaning of LateRatio (ASRatio) is fundamentally different for year 1 compared to year 9 – a value of 1 for these ratios has a fundamentally different meaning for year 1 compared to year 9). The additional analysis is still based on multinomial logistic regression while being deemed inappropriate (if so, what is the use of employing the technique?). It is argued that this is needed to demonstrate the non-linear relationship between the dependent and financial health. Why not using an appropriate estimation technique and including squared financial health as an additional independent? Doing so, you would be able to test the non-linear relationship as well. In sum, I keep having fundamental econometric concerns (although clear guidance was provided in the previous review round).

In addition, I note inaccuracies in reported figures. For example, the last column in Table 7 (Altman scores) should correspond to the figures reported in column ‘Altman’ in Table 6. I note differences for all reported values… How come? Although differences might be considered small, they should not be there. [Same comment for Tables 8 and 9.]

Writing of the paper needs careful attention (and needs to be further improved).

Reviewer 2 Report

Dear authors,
Thank you very much for considering both my and other reviewers' suggestions. In my opinion, the text is much better. However, the work structure is still severely disrupted. First of all, the introduction should be an introduction that outlines the background, identifies research gaps, presents the research objectives and the work structure. As it stands, the introduction has two pages. It has a lot of content that should be found later in the work. Only hypothesis 1 appears in the text. I am asking for an improvement in the structure of the work.

Reviewer 3 Report

Well done!

Author Response

This manuscript is a resubmission of an earlier submission. The following is a list of the peer review reports and author responses from that submission.